# TuneMV3D: Tuning Foundational Image Diffusion Models for Generalizable and Scalable Multiview 3D Generation

## Abstract

Considerable progress has been made in 2D image generation, but 3D content creation lags behind due to a lack of large-scale, high-quality 3D datasets. To mitigate this gap, a recent line of work leverages 2D diffusion models for 3D generation but usually requires object-specific overfitting, making them unscalable. In this paper, we present TuneMV3D, a novel approach to generating diverse and creative 3D content in a scalable feedforward manner. At the core of TuneMV3D, we tune a foundational image diffusion model using a much smaller-scale 3D dataset while utilizing multi-view images to bridge the gap between 2D and 3D. This allows for the direct prediction of consistent, multi-view 3D representations from 2D diffusion models. We design an interactive diffusion scheme that is facilitated by jointly optimized latent SparseNeuS to ensure that the multi-view generations are consistent. Additionally, we propose a consistency-guided sampling strategy that preserves the creativity of the foundational image diffusion model while maintaining multi-view consistency. Using TuneMV3D, we can successfully distill the 3D counterpart of what can be created by a 2D foundation model, thereby generalizing beyond the small 3D tuning set and enabling scalable and diverse 3D content creation. An anonymous website showcasing the results is available at https://tunemv3d.github.io/.

## 1 Introduction

Recently, we have witnessed tremendous progress in foundational image diffusion models. Thanks to the billions of image-text pairs available on the internet, such models can easily generate diverse and visually appealing images from just a short textual prompt. As a comparison, 3D content creation is still in a preliminary stage regarding data diversity and controllability. This is mainly due to the lack of large-scale high-quality 3D datasets. The largest publicly available 3D datasets Deitke et al. (2022) are still of smaller orders of magnitude than existing image datasets Schuhmann et al. (2021). Mitigating this gap and boosting 3D content creation is in urgent need since it would dramatically enrich interactive environments commonly appearing in vision, graphics, and robotics communities.

To break the barrier set by small-scale 3D datasets and allow truly diverse and controllable 3D generation, a recent trend gets rid of the need for 3D datasets and instead focuses on distilling priors from foundational image diffusion models Wang et al. (2022a); Metzer et al. (2022); Poole et al. (2022); Lin et al. (2022). These works usually optimize for a specific 3D object while encouraging its rendering to follow the diffusion prior. However, generating 3D content this way is far from scalable as it requires iterative training and overfitting for each object one by one. We, therefore, ask the question: can we generate diverse 3D content in a feedforward manner while still benefiting from the content creation power of foundational image diffusion models?

One idea would be changing the generation paradigm by tuning foundational image diffusion models using small-scale 3D datasets. This relates to customized image generation Zhang & Agrawala (2023); Mou et al. (2023) whose tuning datasets and generation outcomes both lie in the 2D domain though. Utilizing the customization paradigm for creative 3D generation presents two significant challenges. First and foremost, we need to bridge the 2D-3D gap so that 2D diffusion models can

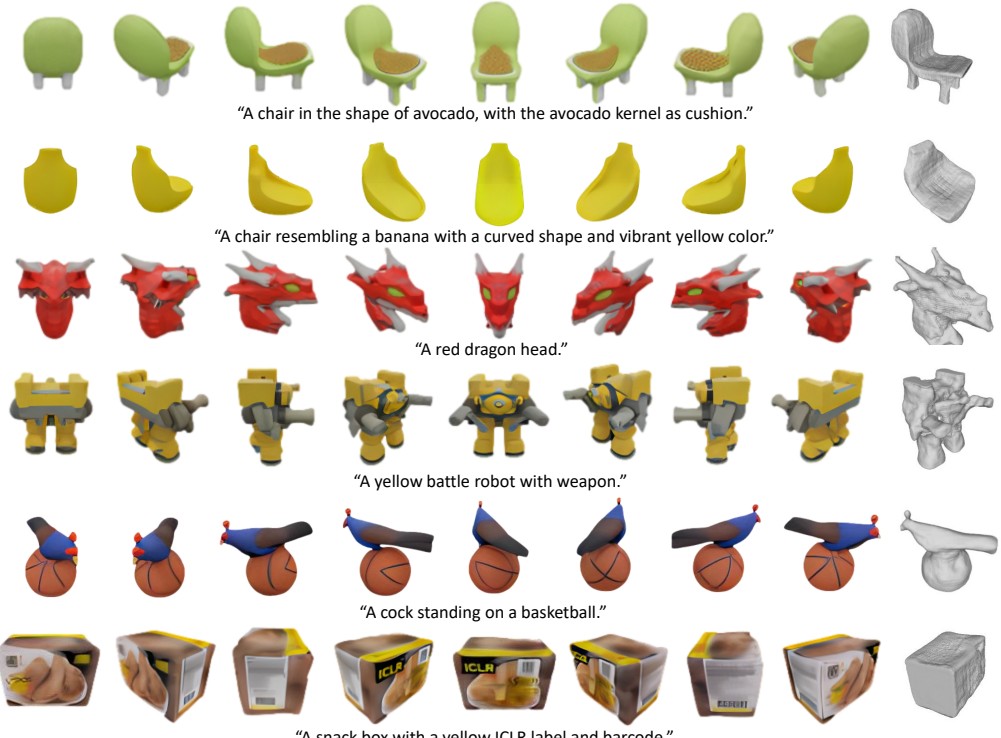

"A chair in the shape of avocado, with the avocado kernel as cushion."

"A chair resembling a banana with a curved shape and vibrant yellow color."

"A red dragon head."

"A yellow battle robot with weapon."

"A cock standing on a basketball."

"A snack box with a yellow ICLR label and barcode."

Figure 1: TuneMV3D can comprehend 3D correspondence and inherit creative knowledge from foundational image diffusion model, requiring only fine-tuning on the ShapeNet-Chair (top two rows) or Objaverse-mini (bottom four rows).

directly predict 3D content. Second, it is crucial to preserve the creative capacity of 2D diffusion models during tuning and avoid overfitting to the small 3D dataset used for customization reference.

To tackle the above challenges, we have two findings. First, multi-view images, a 3D representation being adopted in a wide range of 3D understanding Chen et al. (2017); Wang et al. (2022b) and generations tasks Anciukevičius et al. (2022); Watson et al. (2022); Wen et al. (2019), can naturally bridge 2D and 3D. As long as we can generate multi-view images in a consistent manner, we are able to create 3D content. Second, existing diffusion customization works usually factorize the properties of interest (e.g., style) from object identities. They let the adaptors focus on such properties and still rely on the foundational image diffusion model for object identity creation. This way the creativity of the foundation model can be kept largely. Analogically, while tuning foundational image diffusion models to create multi-view 3D representations, we should design our adaptor to focus on stereo correspondences rather than object identity.

Based on the findings above, we present TuneMV3D, a framework that could tune a foundational image diffusion model using a small-scale 3D dataset without overfitting for feedforward multiview 3D generation. At the core of TuneMV3D is a novel interactive diffusion scheme designed for 3D consistency of the multiple views. Since separately generating individual image views using a diffusion model would easily result in cross-view inconsistency even with the same text prompt input, we propose to interactively conduct multiview diffusion denoising. This involves allowing each pixel in each view to consider relevant pixels from other views before determining the denoising direction. To enable this, we enhance a pre-trained diffusion model with a control branch Zhang & Agrawala (2023) for each view. The control branch incorporates pixel-wise visual cues from other views through centralized and pairwise geometric conditions. The centralized condition is obtained by rendering a low-resolution latent SparseNeuS Long et al. (2022) that aggregates information from the noisy multiview images at each denoising step. The latent SparseNeuS facilitates the establishment of stereo correspondences and provides 3D-consistent conditions for each view. The pairwise condition is obtained through geometry-aware cross attention from another view, complementing the centralized condition when stereo correspondences go wrong. By fine-tuning only the control branch while keeping the pre-trained diffusion model unchanged, we can preserve the creativity of

the 2D foundation model while focusing the tuning on understanding 3D structures. During testing, to further enhance the multiview consistency, we propose a consistency-guided sampling strategy. With our consistent multiview generation approach, we ultimately present a feedforward method to convert the generated views into high-quality 3D meshes.

To verify the effectiveness of TuneMV3D, we tune DeepFloyd-IF with small-scale 3D datasets: ShapeNet-Chair Chang et al. (2015), and mini Objaverse Deitke et al. (2022). Both qualitative and quantitative experiments show that with only small-scale 3D training data, our method achieves as creative but much more consistent multiview generation compared with baseline methods trained with a much larger amount of 3D data. We also demonstrate the flexibility of TuneMV3D when dealing with various input conditions. In summary, our contributions are threefold: 1) a framework that can tune a 2D foundation model for creative and consistent multi-view 3D generation in a scalable feedforward manner; 2) an interactive diffusion scheme based upon centralized and pairwise geometric conditions for multi-view consistent denoising; 3) a general consistency-guided sampling strategy to further improve the 3D view consistency at test time.

## 2 RELATED WORK

### 2.1 3D GENERATIVE MODELS

3D generative modeling has been widely studied with various forms of 3D representations being explored in early work, including voxels Wu et al. (2016); Gadelha et al. (2017); Smith & Meger (2017); Henzler et al. (2019); Lunz et al. (2020), point clouds Achlioptas et al. (2018); Mo et al. (2019); Yang et al. (2019), mesh Zhang et al. (2020); Shen et al. (2021); Gao et al. (2022) and implicit field Chen & Zhang (2019); Mescheder et al. (2019). Recently, with the powerful capabilities demonstrated by diffusion in content generation Nichol et al. (2021); Saharia et al. (2022); Rombach et al. (2022); Ramesh et al. (2022), there has been substantial research into 3D diffusion, which has significantly improved 3D generation. Our main focus is on text-to-3D model generation. The DreamFusion series methods Jain et al. (2022); Wang et al. (2022a); Poole et al. (2022); Lin et al. (2022); Metzer et al. (2022) introduce an SDS loss based on probability density distillation, enabling the use of a 2D diffusion model as a prior for optimization of a parametric image generator. They optimize a NeRF via gradient descent such that its 2D renderings from random angles achieve low loss. These optimization-based methods can realize Zero Shot generation, but each generation usually requires several hours of optimization time. PointE Nichol et al. (2022) generates a single synthetic view using a text-to-image diffusion model, then produces a 3D point cloud using a second diffusion model that conditions on the generated image. However, it employs a non-open source dataset, large-scale dataset for training, and the sparse point clouds it generates are not easily applicable.

### 2.2 NOVEL VIEW SYNTHESIS

Methods for novel view synthesis can provide references for our model design in 3D-aware image generation, especially those based on sparse input view synthesis Yu et al. (2021); Wang et al. (2021); Chen et al. (2021); Kulhánek et al. (2022); Wynn & Turmukhambetov (2023); Yang et al. (2023); Sun et al. (2023). Novel view synthesis based on sparse view input is also a challenging field. Past methods have treated it as a sparse input reconstruction problem, solved by designing a generalizable NeRF. Recently, generative models have enhanced view synthesis models, greatly improving the fidelity of novel views. 3DiM Watson et al. (2022) and Zero-1-to-3 Liu et al. (2023b) rely on the powerful generative capabilities of diffusion itself and attention structures, using input views as conditions to directly generate novel views. RenderDiffusion Anciukevičius et al. (2022) and DiffRF Müller et al. (2022) introduce 3D structure into diffusion, thereby gaining the ability to generate novel view images. Our method differs from these novel view synthesis methods in that we do not rely on any view input, but instead jointly generate all views.

## 3 METHOD

Empowering foundational image diffusion models with scalable 3D generation ability through limited 3D data presents a formidable challenge, primarily stemming from three key issues: i) the vast domain gap between 2D and 3D, ii) the difficulty in enabling 2D foundational image diffusion models to comprehend generic 3D structures and contents, and iii) the challenge of preserving the creativity of the 2D foundation model after adapted with a small 3D dataset.

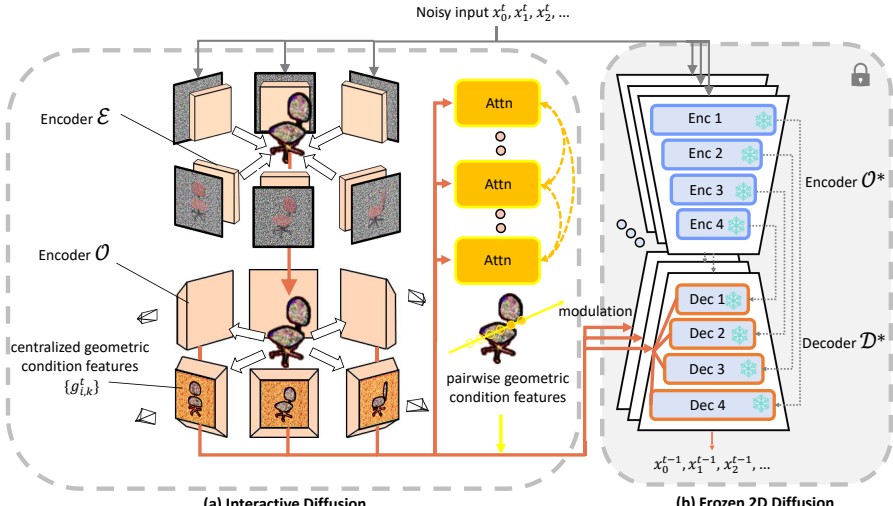

Figure 2: The overall architecture of TuneMV3D. It adopts multi-view as 3D representation and utilizes (a) Interactive Diffusion to exchange information between multiple views and obtain 3D consistent interactive features to modulate (b) the pre-trained 2D foundational image diffusion model.

To tackle these issues, we present TuneMV3D, a scalable 3D generation framework tuned from a 2D foundational image diffusion model, as shown in Fig. 2. First, TuneMV3D adopts consistent multi-view images to bridge 2D and 3D. Specifically, given a text $c$, it jointly generates a set of multi-view images $\mathcal{I}$ as well as a corresponding neural surface field (NeuS) Long et al. (2022) $\mathcal{F}$: $\mathcal{I}, \mathcal{F} = \mathcal{M}(c, \mathcal{O}^*, \mathcal{D}^*)$, where $\mathcal{O}^*$ and $\mathcal{D}^*$ denotes the encoder and decoder of the pre-trained 2D diffusion model (e.g., DeepFloyd-IF, Stable Diffusion Rombach et al. (2022)). $\mathcal{M}$ represents TuneMV3D.

Second, since directly learning generic 3D structure and content from a small-scale 3D dataset is challenging, we factorize 3D generation into two parts i) structured stereo correspondence learning ii) diverse object identity creation. The first part is what the tuning process should focus on and what we need to learn from the small-scale 3D dataset. We propose an **Interactive Diffusion** scheme, which facilitates the exchange of information among multi-views through stereo correspondences. The correspondences are both explicitly established through the multi-view induced latent SparseNeuS and implicitly enforced through a geometry-aware cross attention. The second part is what we should preserve from the pre-trained 2D diffusion model. Inspired by ControlNet Zhang & Agrawala (2023), we freeze the pre-trained 2D diffusion model during tuning and **Modulate** it with the 3D-consistent features derived from the interactive diffusion scheme to fully utilize the content generation capability of the original 2D diffusion model.

Third, we also propose a **Consistency Guidance Sampling** strategy to further improve the consistency of generated multi-view images. It optionally amplifies the influence from the interactive diffusion scheme during some denoise steps to enhance the multi-view consistency flexibly.

Next, we will first introduce the Interactive Diffusion (§3.1) and Multi-view Modulation (§3.2) scheme, and then detail the Consistency Guidance Sampling Strategy (§3.3). In addition, we introduce how to modify the basic text-to-3D TuneMV3D to scalable multi-view to multi-view framework (including single view lifting) in §3.4, and discuss how we efficiently convert the generated views into high-quality 3D meshes in §3.5.

## 3.1 INTERACTIVE DIFFUSION

Multi-view images correspond to a valid 3D object only when they consistently possess structured stereo correspondence. Therefore, it is critical to develop a multi-view diffusion mechanism that guarantees the consistent generation of multi-view images. The forward diffusion pass is relatively easy to design by adding independent noises to every single view. The main challenge is how to make these views interact with each other and maintain a consistent denoising direction. Current multi-view methods typically sidestep this issue by denoising the multi-view images one by one, while conditioning on previously cleaned images. However, such a sequential generation strategy

is sub-optimal and unsuitable in our setting due to the following drawbacks: i) at the start of the sequence, particularly the first one, there is minimal (or no) information from other views, which hinders the adaptor from learning structured stereo correspondence; ii) sequentially denoising views cannot facilitate interactive information exchange between views, implying that the previously generated views cannot be influenced by the later ones, limiting the ability to simultaneously adjust and modulate each single-view 2D diffusion.

Therefore, we utilize a NeuS Long et al. (2022) based module to convert features from other noisy views into the current view, which includes a centralized geometric condition and a pairwise geometric condition mechanism to achieve interactive information exchange among different views. We find that the features from other views, despite their noisy nature, can effectively complement the current view while promoting consistency. Moreover, the NeuS itself can also be gradually denoised as the multi-view diffusion process progresses. This design allows us to obtain both 3D consistent denoised images and an internal NeuS.

Next, we formulate the two core components of our interactive diffusion: i) centralized geometric condition to interact with each view using a unified radiance field, and ii) pairwise geometric condition to interact with each pair of two views to establish more flexible correspondence, handling cases such as occlusion.

### 3.1.1 CENTRALIZED GEOMETRIC CONDITION

Before progressing towards the centralized geometric condition of our interactive diffusion, we first revisit the original 2D diffusion denoising process, serving as a single-view denoiser in our setting. Taking one noisy view $x_i^t$ and a text $c$ as input, where $i$ denote a specific view index and $t$ is the noise level, the foundational image diffusion model utilizes its encoder $\mathcal{O}^*$ to derive hierarchical features $\mathcal{H} = \left\{ h_{i,k}^t \right\}_{k=1}^{L}$ and then use its pyramid decoder $\mathcal{D}^*$ to denoise $x_i^{t-1}$ from $\mathcal{H}$. It's noteworthy that $h_{i,k}^t$ is passed through a residual connection and added to the corresponding $L$ layers in the decoder, as shown in Fig. 2 (b).

Different from the single-view 2D diffusion, we simultaneously take $n$ noisy view $\{x_i^t\}_{i=1}^{n}$ and a text $c$ as input, and first train an image encoder $\mathcal{E}$ to encode each $x_i^t$ into latent features $f_i^t$. Subsequently, we train a Neural Surface Field (NeuS) to aggregate the extracted multi-view features and obtain hierarchical latent feature fields $\left\{\mathcal{F}_k^t\right\}_{k=1}^{L}$ corresponding to $L$ 2D decoder layers $\mathcal{D}^*$ as shown in Fig. 2 (a). Each $\mathcal{F}_k^t$ is expected to predict the $k$-th level centralized geometric condition features when being queried at any input view as follows:

$$g_{i,k}^t = \mathcal{F}_k^t(q_{i,k}, f_i^t), \tag{1}$$

where $g_{i,k}^t$ is the rendered features and $q_{i,k}$ represents the query rays at view $i$ and level $k$.

To be more specific, for a single ray $r_{i,k}(m) = o + md$ from $q_{i,k}$ which extends from the camera origin $o$ along a direction $d$, we first project all the query points along it into every view and gather features using interpolation. Then, we aggregate them by performing an average operation to obtain the feature $\hat{f}^t$ for each 3D query point:

$$\hat{f}^t(r_{i,k}(m)) = \frac{1}{n} \sum_{j=1}^{n} \text{Interpolate}(\pi_j(r_{i,k}(m)), f_j^t), \tag{2}$$

where $\pi_j$ denotes the projection operation from 3D space to view $j$. Then we feed the aggregated features into the following signed distance MLP and feature MLP to get the SDF field $s$ and feature field $c$. The SDF field $s$ is then converted to density field $\sigma$ following Long et al. (2022). Subsequently, we can predict the view-consistent interactive features following the traditional volumetric rendering process Mildenhall et al. (2021):

$$g(r_{i,k}) = \int_0^\infty T(m)\sigma(\hat{f}^t(r_{i,k}(m)), t)c(\hat{f}^t(r_{i,k}(m)), d, t)dm, \tag{3}$$

where $T(m) = \exp(-\int_0^m \sigma(r_{i,k}(s), t)ds)$ handles occlusion. However, we find it difficult for NeuS to directly predict high-dimensional features simultaneously, causing slow convergence speed and huge computational costs. To mitigate this, we change to predict one low-dimensional feature field

$\mathcal{F}^t$ and then utilize another encoder $\mathcal{O}$ to map it into $L$ high-dimension feature spaces. In practice, we let $\mathcal{F}^t$ directly predict the $\boldsymbol{x}_i^0$ and make $\mathcal{O}$ a trainable copy of 2D diffusion encoder $\mathcal{O}^*$.

Upon obtaining the centralized geometric condition features $\left\{ \boldsymbol{g}_{i,k}^t \right\}$, for i = 1, ..., n and k = 1, ..., L, we can control the denoising direction of each view by simply fusing these features with the original feature $\{ \boldsymbol{f}_i^t \}_{i=1}^n$, and then denoise for $\{ \boldsymbol{x}_i^{t-1} \}_{i=1}^n$. However, the centralized geometric condition may force some imperfect correspondence at some times (e.g. occlusion), therefore we further propose the pairwise geometric condition to establish more flexible correspondence between views.

### 3.1.2 PAIRWISE GEOMETRIC CONDITION

We propose a novel geometry-aware attention scheme to produce pairwise geometric conditions. Cross-attention between view images is beneficial for view consistency, which has been proven in recent works Liu et al. (2023b). However, these methods roughly apply global cross attention by attending all pixels of other views for each query pixel, resulting in aimless attention and difficult convergence. We conquer this problem by leveraging 3D geometric priors to locate the candidate keys and values for each query. For every ray cast from a query pixel, we can sample 3D points along the ray and then project them onto all the views, restricting the interested keys to the projected pixels. Moreover, we could roughly locate the query pixel's unprotected 3D positions by our Neural Surface Field to further limit the attention range.

Specifically, given one query pixel $p$, we first find its $N$ neighbor pixels to form a patch and emit a frustum $\mathcal{V} = \{r_i\}_{i=1}^N$ to the 3D space instead of only one ray, establishing more robust correspondence. Then we reuse the sampled locations in our NeuS to introduce geometric priors, which are obtained through an importance sampling strategy. In this way, we can project all the sampled points onto other views and find the candidate keys. Furthermore, the density derived from the sampled 3D points serves as an additional attention weight to facilitate attention training by simply multiplying the original one.

In this way, any two views can establish correspondence through pairwise geometry-aware attention, which allows each single view to attend to information directly from other views, significantly improving the convergence speed and consistency.

### 3.2 MULTI-VIEW MODULATION

To ensure the 2D diffusion prior remains unperturbed and preserves its creativity, particularly when training on small datasets, we fix the weights of the entire 2D diffusion model. Next, we introduce how to incorporate the centralized geometric condition and pairwise geometric conditions into the fixed foundation model.

**Centralized Geometric Condition Modulation**   Given the $i$-th view predicted centralized geometric condition features $\left\{ \boldsymbol{g}_{i,k}^t \right\}_{k=1}^L$ from the NeuS, we carefully use it to modulate the fixed 2D diffusion. Remembering that the fixed 2D diffusion establishes connections $\mathcal{H} = \left\{ \boldsymbol{h}_{i,k}^t \right\}_{k=1}^L$ between its encoder and decoder, inspired by ControlNet Zhang & Agrawala (2023), we apply zero convolutions to each converted feature, subsequently adding these to the initial residuals to incorporate them into the fixed 2D decoder $D^*$: $\hat{\boldsymbol{h}}_{i,k}^t = \boldsymbol{h}_{i,k}^t + \text{ZeroConv}(\boldsymbol{g}_{i,k}^t)$. The zero convolutions are initialized with zero and gradually updated to smoothly modulate the pre-trained 2D diffusion model.

**Pairwise Geometric Condition Modulation**   Recent works Wu et al. (2022) introduce the additional attention module by directly injecting them between the original attention layers of the pre-trained model, tuning the linear projections only. However, we find that such a method inevitably destroys the original priors to some extent. Consequently, given that we only aim to establish multi-view consistent feature residuals, we incorporate our geometry-aware attention module in our trainable encoder $\mathcal{O}$ to understand pairwise correspondence more flexibly and robustly.

### 3.3 CONSISTENCY GUIDANCE SAMPLING

TuneMV3D can efficiently learn structured stereo correspondence while exhibiting impressive creativity from the 2D foundation model due to the above designs. To further control the consistency,

we propose a consistency guidance sampling, which is classifier-free Ho & Salimans (2022) and introduces no additional costs.

The consistency Guidance phase enhances the exchange of interactive information by inserting consistency guidance steps into normal steps uniformly. And we set $K$ to represent the ratio of the number of consistency enhancement steps to the total sampling steps. Then we can optionally enhance the consistency by adjusting ratio $K$. It amplifies the consistent part of the denoising result. To be specific, in each consistency enhancement step, we first input $\boldsymbol{x}_i^t$ directly into our trainable encoder $\mathcal{O}$ to acquire view-independent features $\left\{\boldsymbol{f}_{i,k}^t\right\}_{k=1}^L$, replacing the centralized geometric condition features $\left\{\boldsymbol{g}_{i,k}^t\right\}_{k=1}^L$, then decode the independent denoised result $\overline{\boldsymbol{x}}_i^{t-1}$, a process akin to a dropout operation for the NeuS. At the same time, we leverage the condition features $\left\{\boldsymbol{g}_{i,k}^t\right\}_{k=1}^L$ to obtain $\boldsymbol{x}_i^{t-1}$ as usual. Then we can utilize a consistency guidance scale $\alpha$ (set to 7.5 in our experiment) to encourage the final denoised direction to be more consistent:

$$\hat{\boldsymbol{x}}_i^{t-1} = \boldsymbol{x}_i^{t-1} + \alpha(\boldsymbol{x}_i^{t-1} - \overline{\boldsymbol{x}}_i^{t-1}), \tag{4}$$

where $\hat{\boldsymbol{x}}_i^{t-1}$ denotes the final denoised output after one consistency enhancement step.

### 3.4 SCALABLE VIEW GENERATION

In addition to the text-to-multiview generation, our method also supports the multiview-to-multiview pipeline. In the text-to-multiview pipeline, we synchronously sample all views, meaning that all views always have the same noise timestep throughout the sampling process. In the multiview-to-multiview pipeline, given some reference views, we replace the sampling noisy views in corresponding pose with these clean reference views (can be considered as timestep=0) throughout the sampling process, while other views are still gradually denoised. This approach continuously injects information from the reference view into the sampling of new views, ultimately resulting in a novel view that is consistent with the reference view. When the number of reference views is 1, our method is actually converted to an image-to-multiview generation framework.

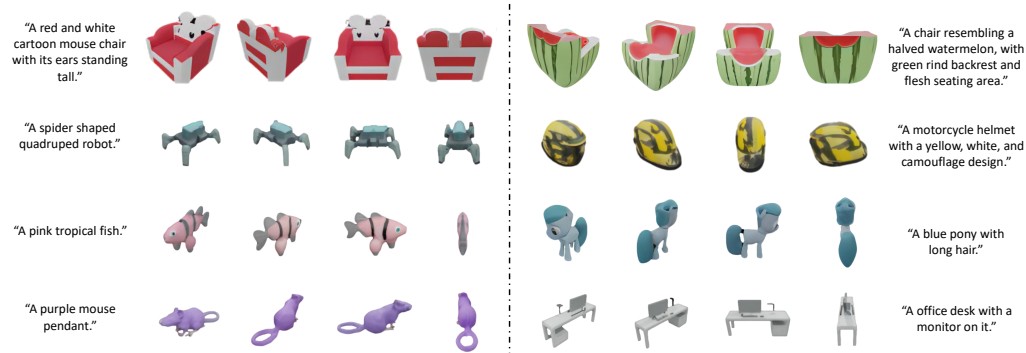

Figure 3: Results of text-to-multiview. The figures from top to bottom represent training by ShapeNet-Chair (1st row) and mini Objaverse, respectively.

### 3.5 POST-PROCESSING

While TuneMV3D can sample images beyond the pre-set views and output a mesh with the help of NeuS, the resolution could be relatively low since the 2D diffusion model we adopted (DeepFloyd-IF, Stable Diffusion) applies the diffusion process in a low resolution. To generate high-resolution images or detailed mesh, we could either use the implicit fields from TuneMV3D as initial weights for the optimization methods (e.g., DreamFusion Poole et al. (2022); Lin et al. (2022)), or utilize mature sparse-view NeRFs. Since our goal is to achieve scalable 3D generation, we employ a variant of SparseNeuS Long et al. (2022) to convert multi-view images into high-resolution Mesh in a feedforward manner. More details and experiment results can be found in our supplementary materials. It's worth noting when combined with optimization methods Poole et al. (2022); Lin et al. (2022), TuneMV3D can not only provide swift previews of results to enhance the efficiency of 3D content creation but also improve performance and convergence efficiency.

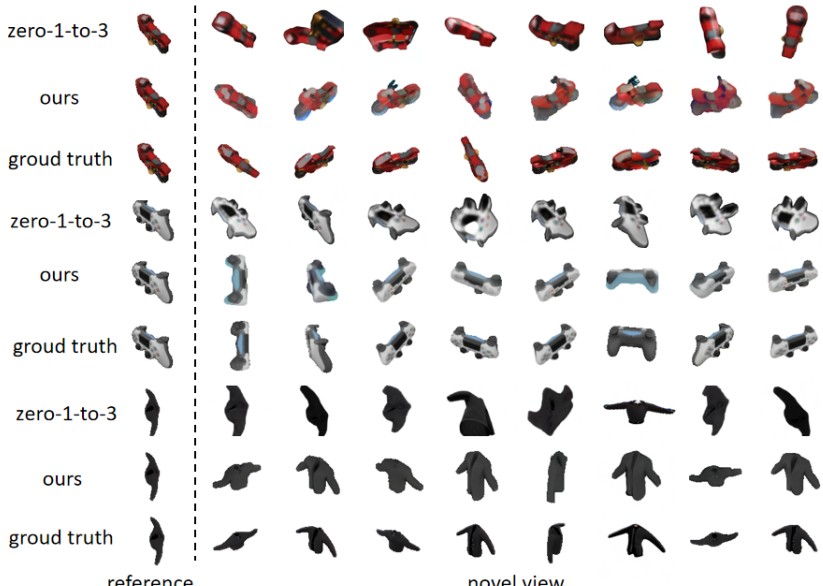

Figure 5: Comparison with other feedforward methods in image-to-multiview.

# 4 EXPERIMENT

## 4.1 DATASET

We assessed the performance of our model in text-to-3D synthesis, employing the representative ShapeNet Chang et al. (2015) chair, comprising 6k 3D chair models. To explore the cross-category generation ability, we also build a mini Objaverse Deitke et al. (2022) by filtering 40k objects within LVIS categories. For each object, we rendered 16 views: eight were used as inputs during training, while the remaining were employed to supervise NeuS's novel view reconstruction, thereby mitigating NeuS degradation. Given the lack of rich text descriptions in 3D data, we employed the ShapNet-Chair text annotations provided by Chen et al. (2019), and generated captions for mini Objaverse using BLIP2 Li et al. (2023).

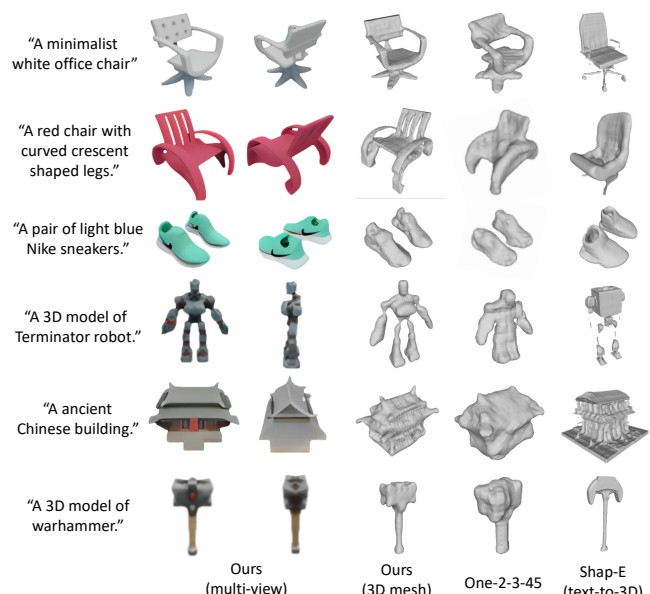

Figure 4: Comparison with other feedforward methods. Our method can generate more creative and precise 3D structures. For One-1-2-3-45, we use the generated view as input and select the best result from all views. For Shap-E, we directly use its text-to-3D pipeline.

## 4.2 IMPLEMENTATION DETAILS

We utilized a pre-trained DeepFloyd-IF-XL as our foundational image diffusion model. In accordance with the ControlNet Zhang & Agrawala (2023) design, the pre-trained weights were fixed while a learnable encoder was copied to furnish the network with additional residual connections. For the NeuS, it runs at $64 \times 64$ resolution. We fine-tune TuneMV3D on ShapNet-Chair and a mini-version of Objaverse and sample 50 DDIM Song et al. (2020) steps during inference.

### 4.3 TEXT-TO-3D RESULTS

The qualitative performance of TuneMV3D is demonstrated in Fig. 1 and Fig. 3, showcasing results on ShapeNet-Chair and Objaverse Deitke et al. (2022). TuneMV3D exhibits an aptitude for sampling objects that align with the style of the dataset, as well as generating objects via free text that deviates far from the training data such as descriptions of chairs with unique shapes and features. Notably, TuneMV3D's results accurately reflect the input descriptions, while preserving a high degree of 3D consistency. This is in line with our primary objective to distill 3D content from the pre-trained 2D model. A modest amount of 3D data is utilized to guide TuneMV3D in acquiring a fundamental comprehension of geometric structure. Then TuneMV3D is capable of harnessing the robust prior knowledge ingrained in the pre-trained 2D diffusion model to facilitate multi-view synthesis. More results can be found on our website.

### 4.4 IMAGE-TO-3D RESULTS

Fig. 4 shows our generation results of lifting a single image to 3D. It can be seen that TuneMV3D can reconstruct both the texture and 3D mesh tightly coherent with the reference view, achieving a sharper geometry and higher reconstruction quality than One-2-3-45 Liu et al. (2023a) and Shap-E Jun & Nichol (2023). Furthermore, We also show single-image to random multi-view images results in Fig. 5. Compared with Zero-123 Liu et al. (2023b), our method has a better comprehension of the multi-view consistency as well as the object identity.

### 4.5 COMPARISON WITH DREAMFUSION

Compared with DreamFusion Poole et al. (2022), while it can achieve zero-shot generation, each generation demands 1-2 hours for 10k step optimization on Tesla A40. In contrast, TuneMV3D exhibits superior efficiency and scalability. We can generate 3D objects in a feed-forward manner within one minute with impressive creativity. Moreover, Dreamfusion suffers from the oversaturation problem, resulting in fake textures. While TuneMV3D can produce realistic and detailed textures due to the well-designed multi-view modulation design.

### 4.6 ABLATION STUDY

We demonstrated the impact of consistency guidance and our choice to implement interactive diffusion. More ablations such as the geometric-aware attention are available in supplementary materials.

**Consistency guidance.** The consistency guidance rate $K$ are utilized to control the consistency extent. The ablation study results of modifying the consistency guidance are displayed in Fig. 6. We find that by employing a larger $K$, the consistency of generated results can be largely improved, which proves the effectiveness of our proposed sampling strategy.

**Interactive Methods.** As introduced in the Section 3.1, there are two ways to obtain 3D consistent

"A Chinese emperor's chair. Elaborately carved with dragons and phoenixes."

Figure 6: Ablation of consistency guidance.

centralized geometric condition features: one is to first render low-dimensional features and then use an additional encoder to map them to hierarchical feature spaces. The other one is to simultaneously predict the high-dimensional features from the NeuS's MLP. In practice, we find that directly predicting high-dimensional features converges more slowly, it does not show obvious consistency after 10k training steps while mapping features from low-dimensional latents achieves consistent generation only after 3k steps, which proved our assumption that establishing a high-dimensional consistent 3D latent field is more challenging. The specific experiment results can be found in the supplementary materials.

## 5 CONCLUSION

In this paper, we propose TuneMV3D, a novel framework to lift foundational image diffusion models for scalable 3D generation, which adopts multi-view representation to bridge 2D and 3D. We propose an interactive diffusion scheme and a consistency guidance sampling strategy. Fine-tuning solely on compact 3D datasets, our approach effectively distills the 2D diffusion priors for scalable 3D generation, charting a promising trajectory in this crucial field.

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
