# Supplementary Material for TuneMV3D: Tuning Foundational Image Diffusion Models for Generalizable and Scalable Multiview 3D Generation

In this document, we provide additional details and insights that further support and expand upon the main contributions of our research, including more details on method implementation (Sec. B.), more experiment details and results (Sec. A., Sec. C., Sec. D.) and the discussion of boarder impacts (Sec. E.). See our project website `https://tunemv3d.github.io/` for the video and more visulization results.

## A.   More Results

To supplement the findings presented in the main paper, we offer extensive visualizations available at Project Page, aiming to provide a more holistic understanding of our methodology. The website features a collection of qualitative results (§4.3), showing the quality and diversity of the 3D content generated by TuneMV3D. In addition, we have incorporated comprehensive 3D representation videos illustrating the effect of post-processing (§3.4). These supplementary resources strive to facilitate a visually enriched, immersive understanding of our research outcomes, thereby augmenting the overall grasp and influence of our work.

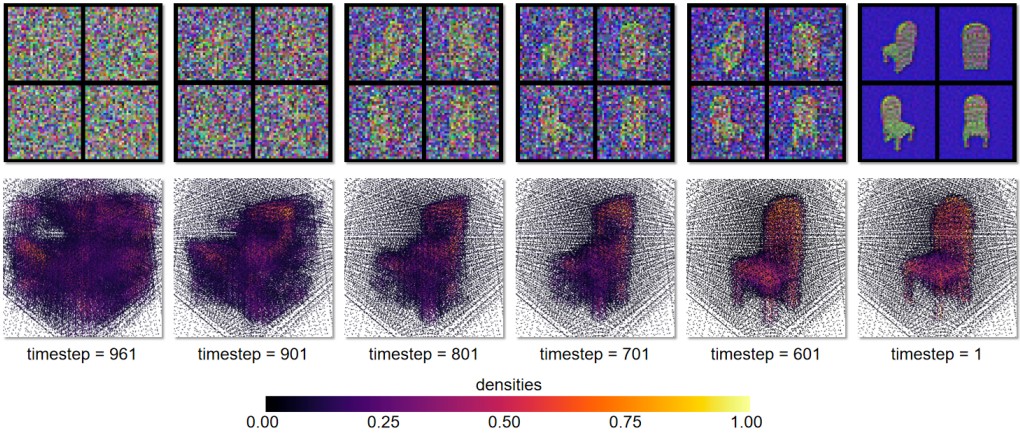

Figure 1: Visualization of intermediate results during sampling via our diffusion model. Top: Visualization of latent denoising. Bottom: Visualization of corresponding NeuS densities.

Submitted to 37th Conference on Neural Information Processing Systems (NeurIPS 2023). Do not distribute.

## B. Implementation Details

This section provides more extensive details about the TuneMV3D architecture, composed of an interactive diffusion scheme and a multi-view modulation module. We first detail the implementation of interactive diffusion and multi-view modulation, then elaborate the detail of our loss functions and training settings.

### B.1  Interactive Diffusion

Our interactive diffusion first utilizes a encoder $\mathcal{E}$ to translate $n$ noisy input views $\left\{\boldsymbol{x}_i^t\right\}_{i=1}^n$ into latent features $\left\{\boldsymbol{f}_i^t\right\}_{i=1}^n$, where $n$ is set to $8$ across all experiments. The encoder, $\mathcal{E}$, employs the architecture of ResBlock [1] and AttentionBlock [6]. It accepts the noisy view $x_i^t$, noise level $t$, and text $c$ as input, incorporates the $t$ embeddings into each ResBlock and exchanges text information by attending the CLIP [5] features that have been extracted from $c$. Subsequent to this, a NeuS is implemented to enable multi-view information interaction. As discussed in the main paper, instead of predicting high dimensional features in one go, we make the NeuS predict the low dimensional $\boldsymbol{x}_i^0$. Specifically, we first apply positional encoding $\gamma$ to the each query point $p$:

$$\gamma(p) = (\sin(2^0\omega p), \cos(2^0\omega p), \sin(2^1\omega p), \cos(2^1\omega p), ..., \sin(2^{M-1}\omega p), \cos(2^{M-1}\omega p)). \tag{1}$$

We adopt $M = 6$ in all experiments and also concatenate the input coordinates $p$ and view directions $d$ along with the encodings. Note that we do not apply positional encoding to the view directions. $\omega$ is a scaling factor, set to $1.5$ for ShapeNet-Chair and $2.0$ for mini-Objaverse respectively.

After aggregating the features for each query point from multi-view features $\left\{\boldsymbol{f}_i^t\right\}_{i=1}^n$ (§3.1), we feed the point encodings and aggregated features into the NeuS to predict $\boldsymbol{x}_i^0$ for each view. Following SparseNeuS, our NeuS network is built upon a series of ResNetFC [1, 7] layers. We also ensure the NeuS is aware of the noise level by inputting $t$ embeddings. Benefit from the $\boldsymbol{x}_i^0$ prediction, we can copy a same encoder $\mathcal{O}$ from the original 2D diffusion to map the low dimensional predictions to hierarchical $L$ features $\left\{\boldsymbol{g}_{i,k}^t\right\}_{k=1}^L$. This encoder, $\mathcal{O}$, mirrors the 2D diffusion encoder $\mathcal{O}^*$ in architecture and initial weights, incorporating four main encoder blocks and one middle block. Note that all the rendered views $x_i^0$ are passed through the same encoder $\mathcal{O}$, therefore the mapped features $\boldsymbol{g}_{i,k}^t$ from $\mathcal{O}$ still maintain 3D consistency. In addition, we also feed the information of raw input $\boldsymbol{x}_i^t$ into $\mathcal{O}$. In more detail, an extra lightweight encoder is applied to $\boldsymbol{x}_i^t$ to extract a noisy feature, which is subsequently added to the initial encoder layer in $\mathcal{O}$. This procedure not only integrates the original noisy information but also facilitates the learning of modulation feature residuals.

Moreover, as shown in Fig. 1, we find that the NeuS progressively reveals a shape that aligns with the multi-view images over the course of the diffusion process, solely supervised by the original single-view image denoising targets. To enhance the quality and convergence speed of the NeuS, we further supervise the rendered prediction with an additional loss, $\mathcal{L}_{neus}$. Specifically, we randomly render extra $n'$ (set to $8$ in our experiment) novel views through the NeuS, together with the $n$ input views, we then apply a $\mathcal{L}_1$ loss to all the $n + n'$ predicted $\boldsymbol{x}_i^0$ and ground truth $\widetilde{\boldsymbol{x}}_i^0$.

### B.2  Multi-view Modulation

As mentioned in the main paper, we apply zero convolutions to interactive features $\left\{\boldsymbol{g}_{i,k}^t\right\}_{k=1}^L$ before integrating them into the decoder. We implement these zero convolutions as 1x1 convolutions, following the methodology of ControlNet [8]. Both the weights and bias of the zero convolution are initialized as zeros, implying that in the initial training step, we have

$$ZeroConv(\boldsymbol{g}_{i,k}^t) = 0, \tag{2}$$

and the overall framework degrades back to a combination of independent single-view 2D diffusion models. As posited by [8], given that the feature $\boldsymbol{g}_{i,k}^t$ is non-zero, the weight and bias of the zero convolution can be optimized into a non-zero matrix in the first gradient descent iteration. Consequently, this approach enables the smooth modulation of the fixed 2D diffusion without drastically disrupting the original 2D priors.

### B.3  Loss Function and Training Details

TuneMV3D is trained end-to-end with each single-view's original denoising loss $\mathcal{L}_{denoise}$ [3] and an additional NeuS loss $\mathcal{L}_{neus}$:

$$L = \lambda_1 \frac{1}{n} \sum_{i=1}^{n} (\mathcal{L}_{denoise}) + \lambda_2 \mathcal{L}_{neus}. \tag{3}$$

At a specific training step with noise level $t$, $\mathcal{L}_{denoise}$ computes the $L_2$ distance between the denoised output $\boldsymbol{x}_i^{t-1}$ from our modulated 2D diffsuion and the ground truth $\widetilde{\boldsymbol{x}}_i^{t-1}$. We then average the losses across all views as our primary loss term. As discussed in B.1, we also supervise the prediction from NeuS with $\mathcal{L}_{neus}$ to improve the NeuS's convergence speed and quality. In our experiments, both $\lambda_1$ and $\lambda_2$ are set to $1.0$.

We fine-tune TuneMV3D on ShapeNet-Chair and mini-Objaverse by AdamW [2] optimizer. We set batch size, learning rate, and weight decay to 4, $5 \times 10^{-5}$, $1 \times 10^{-3}$ for all the datasets. All models are trained on eight NVIDIA Tesla A40 GPUs, each equipped with 46 GB of memory.

## C.  Quantitative Experiment Details

Due to the lack of corresponding ground truth from our text to 3D results, we employ a CLIP R-precision [4] based method to quantitatively evaluate the generation effect, as mentioned in §4.4. We design 30 test prompts for the generative model trained on ShapeNet-Chair, of which 15 prompts are similar to the description of ShapeNet objects, such as "A blue office chair with an adjustable backlog and flip up arms for extra support." The other 15 tend to test the generation capability beyond the training data, such as "A chair that likes avocado, with the brown kernel as its cushion."

For each object, we used CLIP [5] to retrieve the most relevant text from a pool of 100 text candidates for each view. The overall text for the object was determined by selecting the text associated with the view that had the highest number of matching views. In cases where multiple texts had an equal number of match views, we selected the text with the highest CLIP similarity score. Finally, we calculated the R-precision. This metric allows us to quantitatively assess the quality and fidelity of our generated 3D content.

## D.  More Ablations

In the section dedicated to additional ablation studies, we delve deeper into two critical factors as follows:

**The Impact of Raw Noisy Information.**   We analyze the effect of integrating raw noisy information from $\boldsymbol{x}i^t$ into the trainable encoder $\mathcal{O}$, as detailed in section B.1. A comparison of samplings conducted with and without feeding raw noise information under identical training hyper-parameters and steps is presented in Fig. 2 (b) and (c). It is evident that the network convergence slows down when the encoder $\mathcal{O}$ does not receive the original noisy latent image input. However, upon providing raw noisy information, the generated shapes at identical training steps become more diverse and better align with the text prompt. We surmise that the information from $\boldsymbol{x}i^t$ assists the encoder in learning more suitable modulation residuals at the current noise level.

**The Impact of Interactive Methods.**   As mentioned in the main paper, we opted to predict a low dimensional $\boldsymbol{x}_i^0$ then map it to high dimensional features, rather than predicting the high-dimensional features all at once. Fig. 2 (a) and (c) offer a comparison between these two strategies under identical sampling conditions. Our chosen method, depicted in Fig. 2 (c), shows view-consistency in the early 3k training steps and can generate diverse objects after 10k steps. Conversely, the direct prediction of high-dimensional features, shown in Fig. 2 (a), struggles to achieve view-consistency at 10k steps and tends to sample simpler shapes after 20k steps. This comparison underscores the effectiveness of our chosen approach.

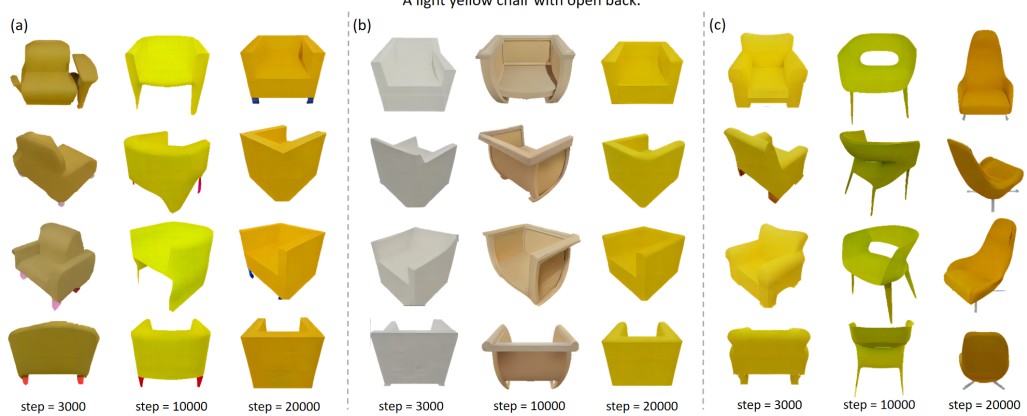

Figure 2: Results for ablation. (a) Results of method for directly predicting high dimensional features. (b) Results without feeding raw information from $x_i^t$ into the trainbale encoder. (c) Baseline results.

## E.   Broader Impacts

The broader impacts of this work are manifold and have implications both within and outside the academic community.

Our work presents the potential to profoundly impact various industries, most notably in computer graphics, gaming, virtual and augmented reality, and robotics. By enhancing the ability of machines to generate 3D content from limited data in a scalable manner, industries that rely heavily on 3D modelling could stand to benefit greatly. For instance, gaming companies could potentially use our technology to quickly generate diverse and realistic 3D environments and characters, thereby reducing development time and costs.

While our research has the potential for positive impact, it's also important to consider possible ethical implications. As 3D content becomes easier to generate and manipulate, issues regarding the misuse of technology and the infringement of intellectual property could arise. Ensuring that this technology is used responsibly and that copyright laws are upheld is essential.