# OpenReview forum: "TuneMV3D: Tuning Foundational Image Diffusion Models for Generalizable and Scalable Multiview 3D Generation"
_ICLR.cc/2024/Conference — ICLR 2024 Conference Withdrawn Submission_

### Official Review · Reviewer_Xd2U · 2023-11-01

**Soundness:** 2 fair
**Presentation:** 2 fair
**Contribution:** 2 fair
**Rating:** 3
**Confidence:** 5

**Summary:**

This paper tackles the problem of text-to-3D generation by lifting a pre-trained 2D diffusion model. It then uses volume rendering / NeuS as a proxy to encourage consistency between generations from various viewpoints. Besides, the authors introduce pairwise geometric conditioning to encourage consistency further. Experiments on the ShapeNet chair and mini Objaverse show improvements on some baselines.

**Strengths:**

The idea of using 3D data to supervise text-to-3D generation is interesting. The 3D generation is quite consistent.

**Weaknesses:**

### 1. Related Works

I think authors miss quite a lot of references/discussions about the recent progress of text-to-3D generation. To just name a few:
- Wang et al., ProlificDreamer. NeurIPS 2023.
- Chen et al., Fantasia3d. ICCV 2023
- Lorraine et al., ATT3D: Amortized Text-to-3D Object Synthesis. ICCV 2023

Arguably, these works all try to tackle the consistency problem (with the help of volume rendering). Please discuss the difference / benefits between this submission and these prior works.

### 2. About Results and Contributions

Can the authors further clarify the reason behind claiming TuneMV3D as a "generalizable" and "scalable" approach?

Though to some extent, I understand that "generalizable" may come from the feed-forward manner. However, on the other hand, the results shown in the paper come from arguably simple objects, e.g., chairs. This quite falls behind other text-to-3D generation qualities in prior works, e.g., ProlificDreamer or ATT3D. Therefore, it is unclear whether this can be claimed as truly "generalizable".

Similarly, "scalable" may also need more justification even considering the training needs 3D datasets, which is less "scalable" to obtain. Prior works only use 2D diffusion models trained on 2D images. Frankly speaking, the whole "interactive diffusion" (Sec. 3.1) is possible because of the existence of 3D data. This somehow undermines the "scalable" statement.

As a natural question, can the authors show some generation results with some texts similar to those used in prior works to showcase the capability of the TuneMV3D?

### 3. Ablation about NeuS Supervision

In Sec. 4.1, the authors state:

> For each object, we rendered 16 views: eight were used as inputs during training, while the remaining were employed to supervise NeuS’s novel view reconstruction, thereby mitigating NeuS degradation.

Can authors provide an ablation about when no NeuS supervision is used? I am quite curious about the role such supervision plays during training. Essentially, I am wondering whether it is due to such supervision that results in consistency.

### 4. Ablation about The Number of Views

Can authors ablate over the number of views used during training? This is related to the bulletin 3 above as this will demonstrate how important the 3D dataset is.

### 5. Appendix Format

The appendix uses NeurIPS's template instead of ICLR's.

**Questions:**

See above.

---

### Official Review · Reviewer_hK4L · 2023-11-01

**Soundness:** 2 fair
**Presentation:** 2 fair
**Contribution:** 2 fair
**Rating:** 3
**Confidence:** 4

**Summary:**

This paper proposes a method to perform multi-view generation from text or image by fine-tuning stable diffusion on a small set of 3D data. The main contribution of the paper is an "interactive diffusion" design combined with SparseNeuS to diffuse multiple views jointly to ensure consistency, as well as a consistency-guided sampling strategy.

**Strengths:**

The proposed method is novel and interesting. Combining SparseNeuS with multiview diffusion by fine-tuning a pretrained diffusion model for the goal of consistent multiview generation makes a lot of sense.

**Weaknesses:**

1. The example generation shown in the paper seems very synthetic. This contracdicts the claim by the author that the method is creative because it’s finetuned on only a small high-quality 3D dataset.
2. For the examples shown in the image-to-3d section, it’s unclear what the sources of these images are.
3. There are no quantitative experiments in the paper/supplementary. The qualitative examples shown in the paper are also not uncurated or randomly selected so likely cherry-picked. These expeirmental studies don't provide enough evidence that the proposed method is superior to priors works in performance.
4. For image to 3D, there’s no comparison with prior methods on any real-world images. Showing some results on real-world images will be necessary in supporting the claim by the authors.
5. Many of the important baselines before the submission of this work are missing in the experiments. For example, DreamFusion, Magic3D, ProlificDreamer, SnapFusion, all of which open-source implementation has been made available. Without both quantitative and qualitative comparisons to these works, it’s hard to evaluate the contribution of the proposed method.
6. One of the main proposed advantages of the method is that the model is able to produce "creative" 3D generation due to the fact that it's only finetuned on a small set of 3D data. However, such "creativity" is not demonstrated in either quanlitative or qualitative experiments. I suggest the authors either remove the claim or provide strong evidence that the proposed model demonstrates such "creativity".

**Questions:**

Overall, the experiment section of the paper is very weak for several reasons:
1. lack of comparison to key baseline
2. lack of any quantitative experiments
3. lack of results on real-world images

---

### Official Review · Reviewer_pX8m · 2023-11-06

**Soundness:** 2 fair
**Presentation:** 2 fair
**Contribution:** 2 fair
**Rating:** 3
**Confidence:** 4

**Summary:**

This paper introduces TuneMV3D, a text-to-3D feedforward generation framework that excels in multi-view image prediction without the need for optimization during testing. TuneMV3D incorporates a latent NeuS for maintaining multi-view consistency and features a new "interactive diffusion module" to leverage 2D diffusion models. The multi-view modulation helps to keep the optimization efficient. Trained on 3D datasets, TuneMV3D produces consistent and reasonable 3D results, also demonstrating its versatility in handling diverse input conditions.

**Strengths:**

There are several notable strengths in this work:

1. The concept of multi-view image generation has garnered significant interest lately, and it appears to be a sensible approach. This is particularly relevant because generating 3D assets directly can be quite challenging with the current available data and algorithms.

2. The incorporation of the frozen 2D diffusion component is a compelling and insightful aspect of this work, with potential practical applications.

3. The generated results are not only promising but also exhibit consistency across different views, indicating the reliability of the approach.

4. The proposed framework seamlessly supports multiple tasks, showcasing its versatility and potential for a wide range of applications.

**Weaknesses:**

**Missing Related Work:**

- The paper lacks mention of "MVDiffusion: Enabling Holistic Multi-view Image Generation with Correspondence-Aware Diffusion" by Tang et al., NeurIPS'23, which focuses on generating multi-view images from text prompts using diffusion models. MVDiffusion incorporates correspondence-aware attention blocks to learn multi-view consistency, which aligns with the pairwise geometric conditions discussed in the submitted paper.

**Clarity of Presentation:**

- The presentation of the paper is somewhat unclear. While it introduces multiple trainable modules, it lacks information about the training loss or the optimization program used. It's crucial to specify whether this framework is trained using SDS loss or follows a supervised learning approach, similar to "zero-123." The statement in the introduction that mentions achieving results "with only small-scale 3D training data" implies supervised learning, but this should be explicitly clarified.

  Furthermore, information (Sec. B.3) that is currently located in the supplementary materials should be incorporated into the main paper. These details, together with necessary loss arguments which are currently missing, are significant and should not be relegated to the supplementary materials. These comments reflect my concerns while reading the main paper, and it's important to ensure that critical information is presented within the main paper, as not all readers may refer to the supplementary materials.

**Issues with Experiments and Evaluation:**

- The experiments and evaluation have room for improvement in several ways:
   1) Qualitative results suffer from very low resolution, falling far short of state-of-the-art optimization-based text-to-3D methods such as "Prolificdreamer."
   2) The results with "zero-123" appear unusual (Fig. 5), possibly due to the low resolution of the reference image. These results are not very convincing. "Zero-123" is typically capable of generating highly detailed images.
   3) The ablation study does not strongly support the claims. Fig. 6 presents only one scene, and the values of K are sparse. The difference between K=0.33 and K=5 is not clearly demonstrated.
   4) The paper lacks quantitative evaluation. While Section 4.4 addresses the image-to-3D task, quantitative benchmarks, such as those used in "zero-123," could provide more rigorous assessment. Similarly, geometry quality evaluation, following the approach in "shape-e," could be valuable.

**Miscellaneous:**

- It's not entirely clear why this paper chooses to train on a subset of Objaverse, limiting itself to objects within LVIS categories. This decision could potentially restrict the impact of the work.
- There is a concurrent work titled "MVDream: Multi-view Diffusion for 3D Generation" by Shi et al. in 2023. While mentioning it might be relevant, it is not an obligation for the authors to do so.
- Failure case and limitation are encouraged to be discussed.

**Questions:**

- **Clarification on Reference View**: In Fig.3, it would be helpful to specify which image serves as the reference view for clarity.

- **Support for Enhanced Efficiency**: In Section 3.5, the paper mentions that TuneMV3D, when combined with optimization methods like Poole et al. (2022) and Lin et al. (2022), can provide swift previews and enhance the efficiency of 3D content creation. However, it would be valuable to provide results or examples that demonstrate how TuneMV3D accomplishes this and improves performance and convergence efficiency.

- **Speed Comparison**: As a feedforward network, one of the key advantages should be speed. It would be beneficial if the authors could report training and inference times/speeds. A comparison between the proposed method and other feedforward methods like shape-e, zero-123, and optimization-based methods such as dreamfusion would provide valuable insights.

- **Rationale for SparseNeuS**: The choice of SparseNeuS is mentioned, but there are no results provided to support this design choice. Including some explanation or results that justify the use of SparseNeuS would add clarity.

- **Lack of Quantitative Experiments**: In the introduction, it's mentioned that both qualitative and quantitative experiments are conducted. However, the location or presentation of the quantitative experiments is not specified. It would be helpful to indicate where the quantitative experiments and their results can be found in the paper.

---

### Official Review · Reviewer_5e7c · 2023-11-08

**Soundness:** 3 good
**Presentation:** 2 fair
**Contribution:** 3 good
**Rating:** 5
**Confidence:** 4

**Summary:**

This paper, TuneMV3D, addresses view-consistent object-level multi-view image generation by harnessing NeuS and ControlNet for latent 3D representation and employing interactive diffusion to facilitate information exchange between multiple views.
Specifically, TuneMV3D necessitates only a small-scale dataset (for instance, of the ShapeNet level) while the foundational diffusion model remains frozen.
Experimental outcomes exhibit that the consistency achieved supersedes that of other counterparts (for example, zero123).

**Strengths:**

- The paper is clearly driven by the goal of enhancing the efficiency and scalability of 3D asset generation. The method delineated herein proficiently truncates the optimization time from a per-scene fitting process to a mere single feed-forward operation.

- The technique proposed in this paper is well-founded, given that the NeuS representation adeptly preserves view consistency within the latent space. The integration of a ControlNet-based modulation imbues the framework with the capability to infuse multi-view details, thereby elevating the fidelity of the synthesized assets.

**Weaknesses:**

- Clarifications Required in Methodology:
In your manuscript, there appears to be some ambiguity: specifically, it remains unclear whether SparseNeuS or NeuS was utilized in your approach.
Could you provide visual evidence to substantiate your assertion that "We find that the features from other views, despite their noisy nature, can effectively complement the current view while promoting consistency"?
Moreover, the mention of "a unified radiance field" necessitates further explanation regarding its application and relevance to your method.

- Evaluation of the Proposed Method’s Effectiveness:
While your study achieves a degree of view consistency in simple object multi-view prediction, it is essential to elucidate the baseline method against which you are comparing. How does the quality compare when you simply integrate the ShapeNet dataset for fine-tuning your baseline method?

- Comparison with Other Methods Needs Expansion:
The analysis presented in your work seems to be limited in scope, offering a comparison solely with zero123 and providing only basic, low-resolution visualizations in Figure 5. For a more comprehensive evaluation, it would be beneficial to include and contrast your approach with other single-image to 3D reconstruction methods cited as references [1,2,3,4].


[1] One-2-3-45: Any Single Image to 3D Mesh in 45 Seconds without Per-Shape Optimization
[2] Make-It-3D: High-Fidelity 3D Creation from A Single Image with Diffusion Prior
[3] NeRDi: Single-View NeRF Synthesis with Language-Guided Diffusion as General Image Priors
[4] Few-View Object Reconstruction with Unknown Categories and Camera Poses

**Questions:**

See the raised questions in weaknesses part.

---

### Author Response · Authors · 2023-11-18

We would like to extend our sincere thanks for the time and effort devoted to reviewing our manuscript. Your comments and suggestions have been helpful in identifying areas for clarification and enhancement in our work. After careful consideration of each point of feedback, and recognizing the potential improvements they propose, after thorough deliberation, we have decided to withdraw our manuscript from consideration for publication in ICLR 2024.

Before finalizing this withdrawal, it's important for us to address some critical points raised during the review. This response is not intended as a defense of our current submission, but rather as a means of clarification for the academic community and for any future research we may conduct in this area. Our goal is to prevent any potential misunderstandings that could arise from the critiques and to contribute constructively to the ongoing discourse in our field.

- **Comparison with related works**:
Our model, TuneMV3D, uniquely generates multiview images and 3D representations simultaneously through feedforward processing. It compares favorably against text-to-3D methods like Shap-E, which yields lower quality outputs, and multiview generation methods like Zero-1-to-3, which struggle with image consistency when camera poses shift largely from the reference image. Compared to the optimization-based method like Prolificdreamer, which produces high-quality results but requires around 10 hours per 3D model and often has the multi-face Janus Problem, TuneMV3D stands out with its capability to generate a model in roughly one minute. This efficiency is a significant advantage of our feedforward, generalizable method over per-shape optimization approaches like Prolificdreamer.

- **Results and Contributions**:
TuneMV3D exemplifies generalizable and scalable.“Generalizable” here refers to its ability to produce diverse content with minimal tuning data. For example, with only 6K objects from ShapeNet-Chair, TuneMV3D can generate a wide variety of chair shapes with different features from free text with great creativity, beyond the synthetic training data. This adaptability allows our model to benefit substantially from advancements in 2D generative models, bridging the gap between 2D and 3D data magnitudes. As is well known, even the largest Objaverse-XL has a significant difference in data volume compared to 2D data. “Scalable” means our model can efficiently generate 3D models in a feedforward manner. Unlike DreamFusion series methods, which require significant time for object generation and lack scalability and practicality, TuneMV3D offers a more efficient solution.